# Variations in echolocation click characteristics of finless porpoise in response to day/night and absence/presence of vessel noise

**Mayu Ogawa**[1,2]*, **Satoko S. Kimura**[1,2,3]

**1** Graduate School of Agriculture, Kyoto University, Kyoto, Japan, **2** Distinguished Doctoral Program of Platforms (WISE), Kyoto University, Kyoto, Japan, **3** Center for Southeast Asian Studies, Kyoto University, Kyoto, Japan

* mayu.ogawa.88@gmail.com

**Data Availability Statement:** All relevant data are within the manuscript and its Supporting Information files.

## Abstract

Small odontocetes produce echolocation clicks to feed and navigate, making it an essential function for their survival. Recently, the effect of vessel noise on small odontocetes behavior has attracted attention owing to increase in vessel activities; however, the effects of the surrounding environmental factor, vessel noise, and day/night on echolocation click characteristics have not been well studied. Here, we examined the effects of vessel noise and day/night on variations in echolocation clicks and click trains parameters. Passive acoustic monitoring of on-axis echolocation clicks produced by free-ranging finless porpoises (*Neophocaena asiaeorientalis sunameri*) was performed at two sites in Japan, Seto Inland Sea and Mikawa Bay, in June–September 2021 and March–August 2022, using A-tag and Sound-Trap 300HF. Generalized Linear Model was used to elucidate the effect of vessel noise, day/night, and surrounding environmental factors (water temperature, synthetic flow velocity, and noise level) on echolocation click and click train parameters. Echolocation click and click train parameters were strongly affected by day/night, whereas the absence/presence vessel noise did not exhibit statistically significant influence. Particularly, -3 dB bandwidth was wider, click duration was shorter, and inter-click intervals in a train were shorter at night, which may facilitate information processing at night, thereby compensating for the lack of visual information. The interaction between day/night and the absence/presence of vessel noise affected the source level of finless porpoises, with higher levels observed in the absence of vessel noise during the daytime compared to other conditions at the site with low vessel traffic. Overall, these results suggest that echolocation clicks by finless porpoise were likely to fluctuate to adapt with surrounding complex environmental conditions, especially day/night.

## Introduction

Small odontocetes produce three types of vocalization sounds: whistles, burst pulses, and echolocation clicks [1]. Whistles and burst pulses are used for communication [2, 3], while

**Funding:** This work was supported by the JST FOREST Program in the form of a grant [JPMJFR2171], the SPIRITS 2020 of Kyoto University, and JSPS KAKENHI in the form of grants [JP19K20460, JP22H05652] to SSK. This study was also supported by The Japan Science Society in the form of a Sasakawa Scientific Research grant [2023-6010], the JST SPRING Program in the form of a grant [JPMJSP2110], and by the Fujiwara Natural History Foundation in the form of funds to MO.

**Competing interests:** The authors have declared that no competing interests exist.

echolocation clicks are used for navigation and feeding [4], making it crucial for their survival. Global vessel noise has increased rapidly over the past several decades [5]. Therefore, the impact of vessel noise on behavior and vocalization in aquatic species [6–12], including small odontocetes [6–9], has attracted increasing attention. Over the years, several studies have been performed to elucidate the impact of vessel noise on echolocation clicks. For instance, Atlantic bottlenose dolphins (*Tursiops truncatus*) and harbor porpoises *(Phocoena phocoena)* decreased emitting buzz rate, feeding trial sound and one of the echolocation clicks [13, 14], in response to the presence of vessels, indicating a decrease in their feeding activity [15, 16]. Additionally, Lahille's bottlenose dolphins (*T. truncatus gephyreus*) emitted significantly fewer echolocation clicks in the presence of vessels in ~ 250-m radius [17]. Moreover, melon-headed whales (*Peponocephala electra*) increased echolocation click source level in correlation with the level of ambient sound pressure [18]. However, previous studies were mainly focused on the effects of vessel noise on the echolocation click train emitting rate, with limited studies on variations in echolocation click and click train parameters like the study of Baumann-Pickering et al. [18]. In contrast, the effects of vessel noise on whistle parameters have been extensively studied. For example, common bottlenose dolphins were observed to shift whistle frequency [19] and increase sound pressure in the presence of vessels [20]. Similarly, studies are necessary to identify echolocation parameters that are affected by vessel noise, especially in small cetaceans to improve conservation efforts.

Previous findings suggest that echolocation click and click train parameters are affected by the surrounding environment [21], and diel changes in echolocation click characteristics have been observed in some species. For instance, melon-headed whales exhibit higher center frequencies at night [18], and harbor porpoises emitted high proportion of click trains with longer inter-click intervals (ICIs) to explore the environment at great distance [22]. Additionally, the diel behavior of harbor porpoises likely depends on changes in the diel behavior of their prey [23–26]. However, studies on the variations in echolocation click characteristics during the day/night have not been conducted in several species, and it is unclear if diel variations in echolocation click are associated with vessel noise. Therefore, it is necessary to examine variations in echolocation click during the day/night and in the absence/presence of vessel noise.

Narrow-ridged finless porpoises (*Neophocaena asiaeorientalis*) are small odontocetes found in the shallow waters of East Asia, consisting of two subspecies, *sunameri* (found in ocean water; hereafter referred to as finless porpoise) and *asiaeorientalis* (found mainly in the Yangtze River; hereafter referred to as Yangtze finless porpoise) [27]. Finless porpoises are often found in shallow sandy areas (< 50 m depth), exposing them to the impacts of anthropogenic activities, such as vessel noise [28]. Both finless porpoise and Yangtze finless porpoise do not emit whistle, but emit narrow-band high frequency echolocation clicks [21, 29–31]. Additionally, Yangtze finless porpoise emits an average of one click train every 5–6 seconds [32–35].

Recently, some studies have examined the impact of vessel noise on Yangtze finless porpoise. For example, increase in vessel noise have been shown to affect Yangtze finless porpoise distribution [36, 37], cortisol levels [38], and activity [39]. Additionally, Yangtze finless porpoise emit more buzz at night for foraging [35, 40], finless porpoises avoid passing boats by diving [41]. However, relatively few studies have investigated changes in echolocation click and click train characteristics for small odontocetes, and no study has been conducted on finless porpoises.

Echolocation click of wild finless porpoise have been reported in the coastal waters of mainland China, Liao-dong-wan Bay in the Bohai Sea, and the western coast of the Taiwan Strait [29, 42]; however, such study has not been performed on wild finless porpoise in Japanese coastal areas. Wild finless porpoises are genetically and morphologically divided into five populations [43–45], each of which is highly vulnerable to anthropogenic impacts due of low level

of genetic diversity [46]. Therefore, understanding the impacts of vessel noise is necessary for the conservation of the finless porpoise in the Japanese coastal areas.

Here, we measured on-axis echolocation clicks and click train parameters of finless porpoises in Seto Inland Sea (St. S) and Mikawa Bay (St. M) in Japan, to elucidate the effects of vessel noise (absence/presence) and day/night on echolocation click and click train parameters. The two recording sites contained genetically and morphologically different local populations of finless porpoises [43–45], with different levels of vessel noise to enable the comparison of changes in echolocation characteristics in response to absence/presence of vessel noise and day/night. It is anticipated that the results of this study would provide valuable information for the conservation of this species.

## Materials and methods

### Study area and recording system

Acoustic recording was conducted at two sites: the Seto Inland Sea (St. S) and Mikawa Bay (St. M), Japan (Fig 1, Table 1), from June–September 2021 and April–August 2022. The St. S finless porpoise belonged to the Inland Sea-Hibiki Nada population, whereas the St. M finless porpoise belonged to the Ise-Mikawa Bay population [43]. Both recording sites had sandy bottom sediments, and fishing boats were the predominant vessels in these areas. A total of 114 and 453 fishing boats were identified in the nearby fishing ports at St. S in 2018 [47] and St. M in 2017 [48], respectively. The data for 2021 and 2022 were unavailable; however, the trend of more vessels at St. M than at St. S was expected to persist. Additionally, there were several ferry routes with more than 30 trips per day within a few hundred meters of the recording site in St. M.

Acoustic and environmental data were recorded at the recording sites, using two A-tags (ML 200-AS8, MMT, Japan), one SoundTrap 300HF (Ocean Instruments, New Zealand), and

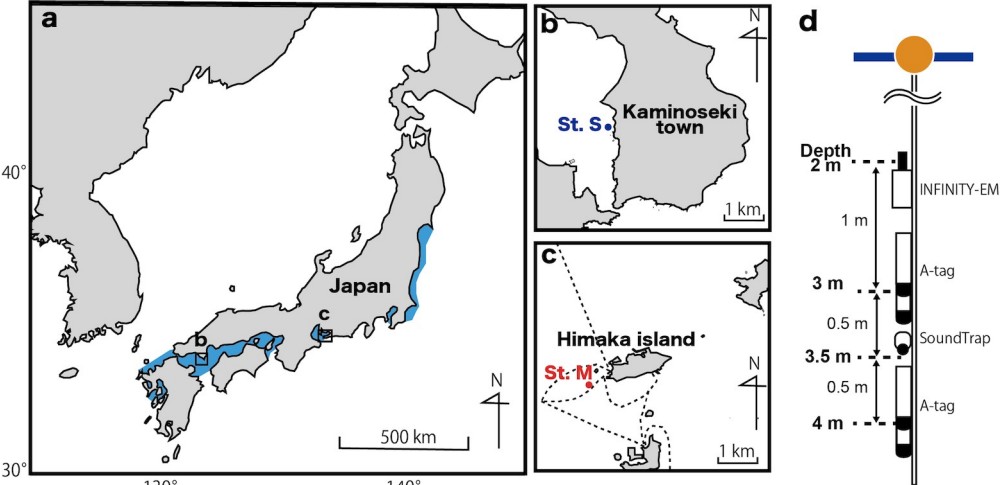

**Fig 1. Recording sites and acoustic array system.** (a) Distribution of finless porpoise in Japanese coastal areas (blue) [45]. This map was modified using data set (https://cyberjapandata.gsi.go.jp/xyz/pale/{z}/{x}/{y}.png) under CC BY 4.0 and QGIS 3.30 (https://qgis.org/it/site/). Shoreline data was derived from: United States. National Imagery and Mapping Agency. "Vector Map Level 0 (VMAP0)." Bethesda, MD: Denver, CO: The Agency; USGS Information Services, 1997. (b, c) Maps of the recording points in Seto Inland Sea (St. S) and Mikawa Bay (St. M), Japan. These maps were modified using data set (https://nlftp.mlit.go.jp/ksj/gml/datalist/KsjTmplt-C23.html) under CC BY 4.0 and QGIS software. The locations of the arrays are depicted by circles, and the dotted lines denote the ferry routes. (d) Configuration of the arrays. From the top, the instruments (black in the figure) were placed with their sensors at 2 m (INFINITY-EM), 3 m (A-tag), 3.5 m (SoundTrap), and 4 m (A-tag) from the water surface (blue line).

one INFINITY-EM current meter (JFE Advantech, Japan). The instruments were assembled on a vertical array, with the INFINITY-EM at a depth of 2 m, A-tag at 3 m, SoundTrap at 3.5 m, and A-tag at 4 m from the surface (Fig 1D). The monitoring arrays were attached to surface buoys, which were anchored and deployed under the supervision of the local fisheries association. Data were obtained by multiple continuous recordings for ~ 8 d. The experimental protocol of this study was non-invasive and was approved by the Animal Experiment Committee of Kyoto University (Inf-K21008).

The A-tag is a stereo event recorder of pulse information capable of recording the time, sound pressure levels of each of the two hydrophones, and the relative azimuth of a pulsed sound that exceeds a set sound pressure threshold. The A-tag was not able to record the frequency information, but the echolocation click train was detected by using smooth changes in sound pressure and the ICI of the consecutive clicks [34, 35, 49]. The two hydrophones of the A-tag had different frequency responses (maximum sensitivities at 130 and 70 kHz) and were placed 190 mm apart. The time difference between the arrival of the sound from the two hydrophones was measured to determine the relative azimuth, and the sound pressure ratio was used to differentiate between Delphinidae and Phocoenidae [50, 51]. The distance to the sound source from the array was calculated using the relative azimuth values and a trigonometric function [52, 53]. The bandpass filter in the A-tags was adjusted to 55–235 kHz, and amplification was modified to + 60 dB. The sampling rate was set to 0.5 ms and the amplitude threshold to 360 counts (approximately 6.3 Pa).

The SoundTrap 300HF was used to obtain sound information, such as the frequency and sound pressure. The SoundTrap 300HF parameters were set as follows: sampling frequency, 576 kHz; high-pass filter, 600 Hz; clip levels, 172 dB re 1 μPa; and self-noise, 37 dB re 1 μPa (> 2 kHz), with 16-bit resolution. The frequency response was flat in the range of 20 Hz –150 kHz, with 174.9, 175, and 176.3 dB re 1 μPa (±3 dB) at St. S and 174.9, 175.9, and 176.3 dB re 1 μPa (±3 dB) at St. M (Table 1).

The INFINITY-EM current meter was used to record the water temperature and tidal current directional velocities. Water temperature and synthetic flow velocity were considered as one of the environmental factors in the study, and water temperature was also used to calculate

**Table 1. Location and device information for acoustic recording of finless porpoise at Seto Inland Sea (St. S) and Mikawa Bay (St. M), Japan, in 2021 and 2022.**

| Recording site | GPS position | Bottom depth | Recording periods | Total recording duration | SoundTrap 300HF sensitivity |
|---|---|---|---|---|---|
| **St. S** | 33˚51.151N, 132˚06.893E | 16 m | July 20, 2021, 00:00 –July 26, 2021, 19:00<br>August 12, 2021, 00:00 –August 18, 2021, 12:30<br>August 31, 2021, 00:00 –September 6, 2021, 15:30<br>April 24, 2022 00:00 –April 30, 2022, 7:00<br>June 6, 2022, 00:00 –June 12, 2022, 12:30<br>July 4, 2022, 00:00 –July 10, 2022, 11:30<br>August 3, 2022, 17:00 –August 10, 2022, 10:00 | 1103 h | July, 2021 –April 2022<br>175.9 dB (± 3 dB)<br>June, 2022<br>174.9 dB (± 3 dB)<br>July, 2022 –August, 2022<br>175.9 (± 3 dB) |
| **St. M** | 34˚41.833N, 136˚59.555E | 15 m | June 16, 2021, 00:00 –June 20, 2021, 20:30<br>July 1, 2021, 00:00 –July 6, 2021, 7:00<br>July 13, 2021, 00:00 –July 18, 2021, 8:00<br>August 7, 2021, 00:00 –August 12, 2021, 8:30<br>August 26, 2021, 00:00 –August 31, 2021, 10:00<br>September 14, 2021, 00:00 –September 19, 2021, 13:00<br>March 21, 2022, 00:00 –March 26, 2022, 17:00<br>April 18, 2022, 00:00 –April 23, 2022, 8:30<br>May 16, 2022, 00:00 –May 21, 2022, 00:00<br>May 23, 2022, 00:00 –May 28, 2022, 5:30<br>June 13, 2022, 00:00 –June 18, 2022, 4:30<br>July 9, 2022, 00:00 –July 14, 2022, 10:00 | 1528.5 h | June, 2021 –May 21, 2022<br>174.9 dB (± 3 dB)<br>May 23, 2022 –May 28, 2022<br>175.9 dB (± 3 dB)<br>June, 2022 –July, 2022<br>176.3 dB (± 3 dB) |

sound velocity. The data were recorded in burst mode (10 times at 0.1-s intervals, once every 5 min).

## Echolocation click and click train analysis

Echolocation click trains of finless porpoise were screened using the A-tag data. Specifically, on-axis sounds were examined using the A-tag and SoundTrap data, and the source parameters of echolocation click and click train were measured using the SoundTrap data. A-tag data were analyzed using Igor Pro 8.03 (WaveMetrics, USA), and the SoundTrap data were analyzed using both Adobe Audition 14.4 (Adobe, USA) and MATLAB R2021a (MathWorks, USA).

At first, the A-tag data were filtered based on the following parameters to detect echolocation click trains of the finless porpoise: a series of clicks with a duration of 1–200 ms and consisting of a minimum of six pulses were categorized as a click train, while reflected waves (i.e. waves within < 1 ms interval of the previous wave) were excluded [34]. Following Kameyama et al. [50] and Kimura et al. [51], the sound pressure ratio of the two hydrophones in the A-tag was set to ≥ 0.6 to enable the detection of the click trains of most of Phocoenidae, including the finless porpoise. Each click train detected was visually inspected to confirm whether it met the criteria for a click train, using Igor software, according to the procedures described by Akamatsu et al. [33, 34] and Kimura et al. [49]. The number of click trains (on-axis and off-axis) produced by the finless porpoise was calculated based on the data obtained from the A-tag placed at a 4-m depth. Five consecutive clicks with ICI < 10 ms was defined as buzz (feeding sound) [13, 26], and the number of buzz and the ratio of buzzes were calculated. Additionally, click trains and buzzes were categorized as day/night [54, 55], and their hourly average was calculated.

Subsequently, the click arrival time recorded by the A-tag were used as reference to filter the click trains from the SoundTrap data using Adobe Audition. Based on the previous studies [56–58], six criteria were defined to extract on-axis click or click train from the A-tag and SoundTrap 300HF data: (1) click trains must be measured by all five hydrophones in two A-tags and one SoundTrap 300HF; (2) the relative azimuth must be within –35.5˚ to 35.5˚; (3) the amplitude of the series of click trains should first increase and then decrease (*sensu* [59]); (4) the maximum amplitude of the click must be higher than the reflected wave from the sea surface or the seabed when present; (5) the on-axis sound should not be distorted; and (6) the distance between the estimated source and the acoustic array should be within 60 m. Clicks with the highest amplitude in the series of click trains were analyzed. Analysis was conducted only when the signal-to-noise ratio (peak-to-peak amplitude) of the series of click trains was > 20 dB. Criterion (1) was analyzed using the A-tag and SoundTrap data, (2) and (6) were analyzed using the A-tag data, and (3), (4), and (5) were analyzed using the SoundTrap data. The distance between the finless porpoise and acoustic array was estimated using the equation provided by Kimura et al. [52, 53].

Click and click train parameters for on-axis echolocation click of finless porpoises were calculated using MATLAB. Click parameters included the apparent source level (ASL) in dB re 1 μPa, peak frequency and center frequency in kHz, -3 dB bandwidth (BW) in kHz, click duration in μs, as well as click train parameters such as ICI in milliseconds (ms) and the number of clicks per train. ASL was calculated using Eq (1) by Møhl et al. [60]:

$$ASL = RL + TL \tag{1}$$

where *RL* is the received level and *TL* is the transmission loss. *TL* was calculated using Eq (2):

$$TL = 20\log r + r\alpha \tag{2}$$

where $r$ is the estimated distance from the finless porpoise to the acoustic array and $\alpha$ is calculated using the Francois and Garrison equation [61]. Water temperature data used for $\alpha$ calculation was obtained using INFINITY-EM. Salinity data at St. S were obtained from the Hirae Iwakuni fixed-line survey conducted by the Inland Sea Research Department of the Yamaguchi Prefectural Fisheries Research Center at 33°50'12″N, 132°02'04″E (~ 7.7 km from St. S) once a month during the same period, using the Yamaguchi Prefecture submarine and fisheries research vessel "Seto" [62]. Salinity data were measured at St. M at a 3.5 m depth, using the No. 2 Automatic Oceanographic Observation Buoy in Mikawa Bay (34°44'42″N, 137°04'19″E; ~ 9 km from St. M), which is owned by the Aichi Fisheries Research Institute [63]. When data could not be obtained due to equipment maintenance on the buoy, the salinity data obtained at the closest observation time were used. The $\alpha$ (0.048 ± 0.005 dB/m; average ± SD) was calculated using variable water temperature and salinity values. The frequency of the maximum amplitude was defined as the peak frequency, the average of frequencies at -3 dB from the amplitude of the peak frequency was defined as the center frequency, and the frequency bandwidth was defined as -3 dB BW. The click duration referred to the time from when the amplitude of the click became larger than the background noise to when it became smaller than the background noise. ICI was calculated as the average interval between two ICIs before and after the click with the maximum amplitude. Finally, the number of clicks per train was measured.

## Environmental data analysis

The absence/presence of vessel noise was determined by manually listening to the sounds 1 min before and after on-axis click train detection. Additionally, the detected click trains were categorized into day or night based on the time of detection and sunset information at each site. Data from timeanddata.com [54, 55] were used as a reference for sunset and sunrise information; "Yamaguchi" and "Aichi" were used as the reference points for St. S and St. M, respectively. Although some diel behavioral studies on odontocetes used civil twilight times in their analyses [64], we used day and night categories (based on sunset and sunrise), due to the small sample size. The time from sunrise to sunset was defined as day, and the time from sunset to sunrise was defined as night. Water temperature data and synthetic flow velocity were obtained from INFINITY-EM data averaged over 10 recordings at 0.1-s intervals, once every 5 min. The root mean square (rms) amplitudes at 1 min before and after each click train were calculated using the rms function in MATLAB and defined as the noise level (dB re 1 μPa).

## Statistical analysis

All statistical analyses were performed using the R software (version 3.6.3) [65]. The environmental parameters (temperature, synthetic flow velocity, and noise level) at each site were compared using Mann–Whitney U-test. Similarly, click and click train parameters at each site were compared using Student's t-test or Mann–Whitney U-test. Generalized Linear Models (GLMs) with logit link function were generated to analyze the relationship between each parameter and absence/presence of vessel noise, day/night, environmental parameters, and recording sites using lme4 package [66] and lmerTest [67] in R. Click and click train parameters were the response variables; absence/presence of vessel noise (factor type), day/night (factor type), interaction between absence/presence of vessel noise and day/night, temperature, synthetic flow velocity, noise level, recording sites (factor type) were the explanatory variables. ASL was analyzed using GLM with gaussian family and other parameters were analyzed using GLMs with gamma family. The full model was generated, and the best model was estimated with respect to the Akaike's Information Criterion (AIC) using the dredge function included in the MuMIn package [68].

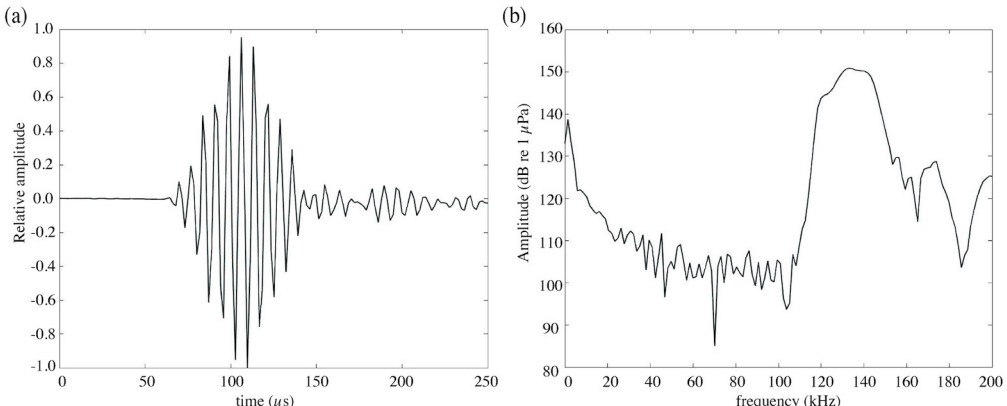

**Fig 2. Typical echolocation click of finless porpoise.** (a) Waveforms and (b) signal frequency. Both (a) and (b) are sounds obtained from the Seto Inland Sea (St. S) data.

## Results

### Echolocation click and click train parameters

A total of 136 and 180 on-axis echolocation click trains of Japanese coastal finless porpoise were detected at St. S and St. M, respectively (Fig 2, Tables 2 and 3). Specifically, 75 and 46 click trains were detected at St. S and St. M during the daytime, while 61 and 134 click trains were detected during nighttime (Table 2). Additionally, at St. S and St. M, vessel noise was detected before or after 80 and 66 click trains, while vessel noise was not detected 56 and 114 click trains (Table 2).

A total of 1,257 on- and off-axis click trains (excluding buzz) were recorded at St. S (810 and 447 at day and night, respectively), whereas 4,104 click trains were recorded at St. M (925 and 3,179 at day and night, respectively). Additionally, 110 (81 and 29 at day and night, respectively) and 120 buzzes (38 and 82 buzzes at day and night, respectively) were recorded at St. S and St. M, respectively. Moreover, the buzz ratios (percentage of total buzz to the total click train) at St. S and St. M were 8.8% (10.0% and 6.5% at day and night, respectively) and 2.9% (4.1% and 2.6% at day and night, respectively), respectively.

Furthermore, the average ASL and peak frequency at both sites were 174 ± 10 dB re 1 μPa (average ± SD) (n = 316; 172 ± 11 at St. S, n = 136 and 175 ± 9 dB re 1 μPa St. M, n = 180; Table 3), and 134 ± 6 kHz (n = 316; 134 ± 6 at St. S, n = 136 and 134 ± 6 kHz St. M, n = 180; Table 3). There were significant differences in temperature, synthetic flow velocity, noise level, ASL, -3 dB BW, click duration, and number of clicks per train between St. S and St. M; however, there was no significant difference in range from array, peak frequency, center frequency, and ICI and between both sites (Table 3).

### GLM model selection

For ASL, the best model was the one with the absence/presence of vessel noise ($p = 0.94$, S1 Table), day/night ($p < 0.01$, S1 Table), interaction between the absence/presence of vessel

**Table 2. Number of echolocation click trains at Seto Inland Sea (St. S) and Mikawa Bay (St. M) during day/night in the absence/presence of vessel noise.**

| | St. S (n = 136) | | St. M (n = 180) | |
|---|---|---|---|---|
| **Vessel condition** | **Absence** | **Presence** | **Absence** | **Presence** |
| Day | 31 | 44 | 8 | 38 |
| Night | 49 | 12 | 58 | 76 |

**Table 3. Summary of environmental parameters, finless porpoise echolocation click and click train parameters at Seto Inland Sea (St. S) and Mikawa Bay (St. M).**
The summary shows environmental parameters (temperature, synthetic flow velocity, and noise level), echolocation click parameters (ASL, peak frequency, center frequency, -3 dB BW, and click duration), and click train parameters (ICI, number of clicks per train, and range from array). ASL was compared using Student's t-test, while the other parameters were compared using Mann–Whitney U-test to determine significant differences between the sites. $0.01 \leq p < 0.05$ is denoted as > or < and $p < 0.01$ is denoted as >> or <<; > and >> indicate that the parameter in St. S were larger than that at St. M, while < and << indicate that the parameters in St. M were larger than those in St. S.

| | Total (n = 316) | | St. S (n = 136) | | | St. M (n = 180) | |
|---|---|---|---|---|---|---|---|
| **Source parameter** | **Average ± SD** | **Range** | **Average ± SD** | **Range** | **p** | **Average ± SD** | **Range** |
| range from array (m) | 17±13 | 2–59 | 18±13 | 3–56 | 0.94 | 17±12 | 2–59 |
| temperature (˚C) | 21±5 | 12–29 | 25±2 | 22–28 | < | 30±3 | 13–32 |
| synthetic flow velocity (cm/sec) | 21±18 | 1–60 | 5±3 | 1–14 | << | 34±15 | 1–60 |
| noise level (dB$_{rms}$ re 1 µPa) | 117±2 | 112–124 | 116±1 | 112–121 | << | 118±2 | 114–124 |
| ASL (dB re 1 µPa) | 174±10 | 137–198 | 172±11 | 137–198 | << | 175±9 | 152–197 |
| peak frequency (kHz) | 134±6 | 117–149 | 134±6 | 117–148 | 0.97 | 134±6 | 120–149 |
| center frequency (kHz) | 134±4 | 119–152 | 134±4 | 119–144 | 0.77 | 134±5 | 120–152 |
| -3 dB BW (kHz) | 25±9 | 7–68 | 22±7 | 7–37 | << | 28±10 | 11–68 |
| click duration (µs) | 65±15 | 38–116 | 73±15 | 45–116 | >> | 59±12 | 38–104 |
| ICI (ms) | 41±22 | 5–162 | 45±25 | 5–162 | 0.05 | 37±19 | 5–144 |
| number of clicks per train | 21±17 | 5–137 | 24±16 | 6–80 | >> | 19±18 | 5–137 |

Abbreviations: ASL, apparent source level; ICI, inter-click interval; BW, bandwidth.

noise and day/night ($p < 0.01$, S1 Table), temperature ($p < 0.01$, S1 Table), and sites ($p < 0.01$, S1 Table) as the explanatory variables (AIC = 2327.8, null deviance = 31291, residual deviance = 27969, Table 4). ASL was higher during the day in the absence of vessel noise at St. S (Fig 3A).

For peak frequency, the null model using a GLM (AIC = 2050.4, null deviance = 0.67, Table 4) was selected as the best model. None of the explanatory variables had a significant effect on peak frequency.

For center frequency, the best model was the one with the absence/presence of vessel noise ($p = 0.12$, S1 Table), day/night ($p = 0.10$, S1 Table), temperature ($p < 0.01$, S1 Table), and noise level ($p = 0.03$, S1 Table) as the explanatory variables (AIC = 1830.0, null deviance = 0.35, residual deviance = 0.32, Table 4), performed using GLM. The absence/presence of vessel noise and day/night did not affect the center frequency.

For -3 dB BW, the best model was the one with the day/night ($p < 0.01$, S1 Table), temperature ($p < 0.01$, S1 Table), noise level ($p = 0.02$, S1 Table), and site ($p < 0.01$, S1 Table) as the explanatory variables (AIC = 2139.6, null deviance = 38.54, residual deviance = 27.10, Table 4). A wider -3 dB BW was observed during nighttime at both sites (Fig 3B).

For click duration, the best model was the one with the day/night ($p < 0.01$, S1 Table), temperature ($p = 0.13$, S1 Table), noise level ($p = 0.15$, S1 Table), and site ($p < 0.01$, S1 Table) as the explanatory variables (AIC = 2458.8, null deviance = 15.31, residual deviance = 10.50, Table 4). Click duration was significantly shorter at night in both sites (Fig 3C).

For ICI, the best model was the one with the day/night ($p < 0.01$, S1 Table) and noise level ($p < 0.01$, S1 Table) as the explanatory variables (AIC = 2725.3, null deviance = 89.57, residual deviance = 77.79, Table 4). ICI was significantly shorter at night at both sites (Fig 3D).

For the number of clicks per train, the best model was the one with the day/night ($p = 0.17$, S1 Table), temperature ($p < 0.01$, S1 Table), and noise level ($p = 0.01$, S1 Table) as the explanatory variables (AIC = 2454.4, null deviance = 145.72, residual deviance = 136.08, Table 4). The absence/presence of vessel noise and day/night did not significantly affect the number of clicks per train.

**Table 4. Selection of the best model generalized linear models (GLM) for echolocation click and click train parameters.** The first column shows the ranks, numbered from the smallest Akaike's Information Criterion (AIC). This table shows models with AICs < 2. The second column shows the explanatory variables for GLMs (vessel = absence/presence of vessel noise, DN = day/night, vessel: DN = interaction between day/night and absence/presence of vessel noise, temp = temperature, flow = synthetic flow velocity, noise level = noise level, site = recording site). Bold text indicates the respective response variable. *df*: degrees of freedom.

| Rank | Model | AIC | ΔAIC | *LogLik* | *df* | weight |
|------|-------|-----|------|----------|------|--------|
| | **ASL** | | | | | |
| 1 | vessel + DN + vessel :DN + temp + site | 2327.8 | 0.00 | -1156.718 | 7 | 0.341 |
| 2 | vessel + DN + vessel :DN + temp + flow + site | 2327.9 | 0.07 | -1155.700 | 8 | 0.330 |
| 3 | vessel + DN + vessel :DN + temp + noise level + site | 2329.7 | 1.86 | -1156.594 | 8 | 0.135 |
| | vessel + DN + vessel :DN + temp + flow + noise level + site | 2329.7 | 1.94 | -1155.577 | 9 | 0.129 |
| | **peak frequency** | | | | | |
| 1 | null | 2050.4 | 0.00 | -1023.171 | 2 | 0.072 |
| 2 | DN | 2050.8 | 0.44 | -1022.372 | 3 | 0.058 |
| 3 | temp | 2051.5 | 1.13 | -1022.717 | 3 | 0.041 |
| 4 | flow | 2051.8 | 1.43 | -1022.868 | 3 | 0.035 |
| 5 | DN + flow | 2051.8 | 1.45 | -1021.852 | 4 | 0.035 |
| 6 | vessel | 2052.1 | 1.75 | -1023.028 | 3 | 0.030 |
| 7 | DN + temp | 2052.2 | 1.78 | -1022.016 | 4 | 0.030 |
| 8 | vessel + DN | 2052.2 | 1.79 | -1022.021 | 4 | 0.029 |
| 9 | noise level | 2052.2 | 1.83 | -1023.068 | 3 | 0.029 |
| 10 | site | 2052.3 | 1.96 | -1023.134 | 3 | 0.027 |
| | **center frequency** | | | | | |
| 1 | vessel + DN + temp + noise level | 1830.0 | 0.00 | -908.874 | 6 | 0.104 |
| 2 | DN +temp + noise level | 1830.4 | 0.37 | -910.096 | 5 | 0.086 |
| 3 | vessel + temp + noise level | 1830.7 | 0.64 | -910.233 | 5 | 0.075 |
| 4 | DN + temp | 1831.3 | 1.24 | -911.567 | 4 | 0.056 |
| 5 | DN + vessel + temp + noise level + flow | 1831.3 | 1.30 | -908.477 | 7 | 0.054 |
| 6 | DN + temp + noise level + flow | 1831.4 | 1.37 | -909.560 | 6 | 0.052 |
| 7 | DN + vessel + temp + noise level + site | 1831.9 | 1.89 | -908.772 | 7 | 0.040 |
| | **-3 dB BW** | | | | | |
| 1 | DN + temp + noise level + site | 2139.6 | 0.00 | -1063.659 | 6 | 0.239 |
| 2 | DN + temp + noise level + flow + site | 2140.9 | 1.31 | -1063.270 | 7 | 0.124 |
| 3 | DN + temp + flow + noise level | 2140.9 | 1.34 | -1064.329 | 6 | 0.122 |
| 4 | vessel + DN + temp + noise level + site | 2141.6 | 1.97 | -1063.599 | 7 | 0.089 |
| | **click duration** | | | | | |
| 1 | DN + temp + noise level + site | 2458.8 | 0.00 | -1223.286 | 6 | 0.145 |
| 2 | DN + temp + site | 2459.0 | 0.18 | -1224.416 | 5 | 0.133 |
| 3 | DN + noise level + site | 2459.2 | 0.37 | -1224.509 | 5 | 0.121 |
| 4 | DN + sIte | 2459.9 | 1.06 | -1225.888 | 4 | 0.085 |
| 5 | DN + vessel + temp + site | 2460.7 | 1.90 | -1224.236 | 6 | 0.056 |
| | **ICI** | | | | | |
| 1 | DN + noise level | 2725.3 | 0.00 | -1358.583 | 4 | 0.196 |
| 2 | DN + flow + noise level | 2726.3 | 1.01 | -1358.057 | 5 | 0.118 |
| 3 | DN + temp + noise level | 2726.9 | 1.59 | -1358.348 | 5 | 0.088 |
| 4 | vessel + DN + noise level | 2727.0 | 1.74 | -1358.422 | 5 | 0.082 |
| 5 | DN + noise level + site | 2727.1 | 1.82 | -1358.459 | 5 | 0.079 |
| | **number of clicks per train** | | | | | |
| 1 | DN + temp + noise level | 2454.5 | 0.00 | -1222.167 | 5 | 0.139 |
| 2 | temp + noise level | 2455.1 | 0.60 | -1223.502 | 4 | 0.103 |
| 3 | temp + noise level + site | 2455.4 | 0.89 | -1222.615 | 5 | 0.089 |

*(Continued)*

**Table 4.** (Continued)

| Rank | Model | AIC | ΔAIC | *LogLik* | *df* | weight |
|------|-------|-----|------|----------|------|--------|
| 4 | DN + temp + noise level + site | 2456.2 | 1.65 | -1221.952 | 6 | 0.061 |
| 5 | vessel + DN + vessel :DN + temp + noise level | 2456.5 | 1.97 | -1221.065 | 7 | 0.052 |
| 6 | DN + temp + flow + noise level | 2456.5 | 1.98 | -1222.119 | 6 | 0.052 |

## Discussion

### Effects of presence of vessel noise on echolocation click and click train parameters

ASL was higher during the day in the absence of vessel noise at St. S than under other conditions, whereas ASL decreased in the presence of vessel noise, which was contrary to previous findings. Most previous studies showed an increase in echolocation click source level in beluga (*Delphinapterus leucas*) in the presence of ferry [69], and melon-headed whales increased echolocation click source level with increasing ambient noise regardless of day and night [18]. The phenomenon of increased vocalizations in the presence of noise is known as the Lombard effect [70], which has been observed in cetaceans [20, 71] as well as in birds and bats [72],

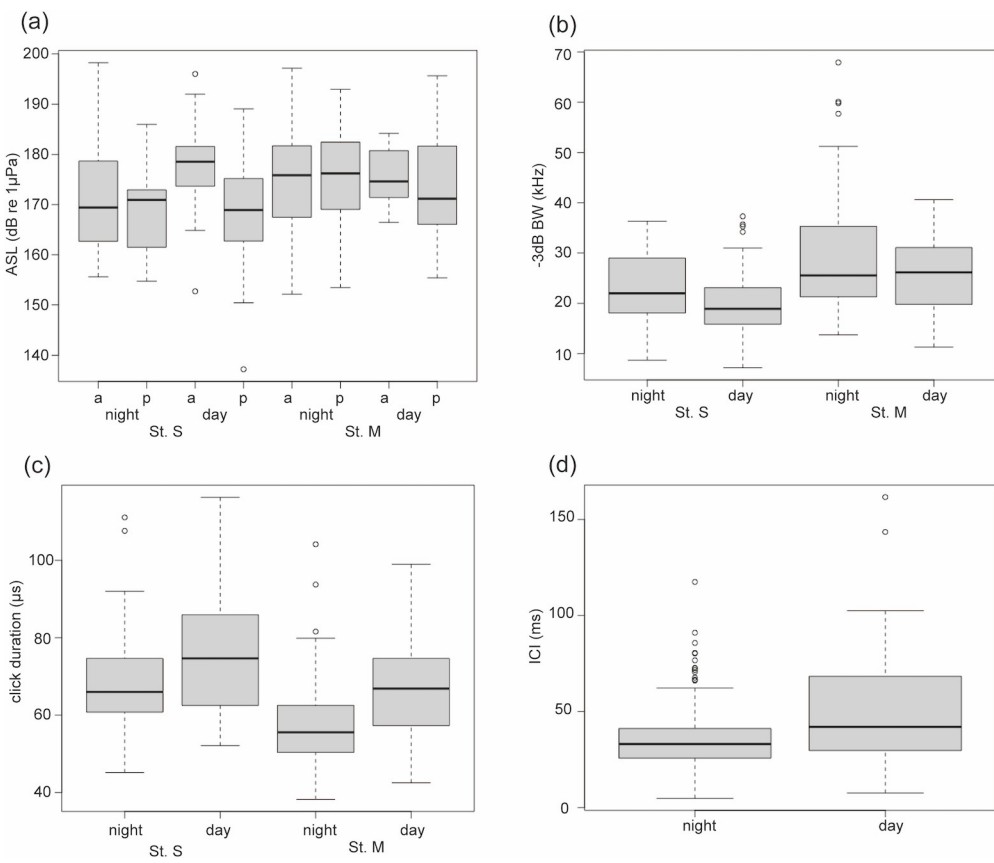

**Fig 3. Box plots for the relationship between the explanatory variables and respective response variables.** The box plots represent (a) ASL, (b) -3 dB BW, (c) click duration, (d) ICI. The lower box limit represents the 1st quartile and the top box limit represents the 3rd quartile. The middle line in the box represents the median. Refer to Table 3 for sample size. In the (a), "a" means absence of vessel noise, and "p" means presence of vessel noise.

indicating a common response to noise. However, contrary to this pattern, the finless porpoises in this study exhibit a decrease in sound pressure under vessel noise. Some studies showed that the vocalization rate of echolocation clicks decreases in the presence of vessels due to acoustic interference and enhanced vigilance [15, 73]. Moreover, cetaceans have been shown to exhibit similar responses to predators and anthropogenic noise [74]. Therefore, it is possible that the observed decrease in ASL in response to vessel noise in the present study may be attributed to vigilance behavior in the presence of vessels. The solo impact of vessel noise on ASL was not significant; instead, distinct influences were observed with the interaction between vessel noise and day/night. Finless porpoise may be more cautious at night than during the day owing to shorter visual range at night. On the other hand, the decrease in ASL may lead to a reduction in search range, potentially leading to a decline of feeding opportunities.

ASL showed different changes between the two recording sites in response to factors of day/night variations and the presence of vessels. At St. S, ASL was affected by the absence/presence of vessel noise and day/night, with a notably higher ASL during the daytime in the absence of vessel noise. In contrast, only slight changes were observed in ASL between the day/night or the absence/presence of vessel noise at St. M. The differences in response between the recording sites might be due to various factors, such as habituation to vessel noise, prey species, and other complex environmental factors. For instance, Indo-pacific bottlenose dolphins in areas with low vessel noise have been shown to respond more strongly to vessels than dolphins in areas with high vessel noise [75]. In the present study, variation in ASL was low at the site with high vessel noise (St. M), whereas variations in ASL was high at the site with low noise (St. S). Therefore, the variation in ASL in finless porpoises could be attributed to different responses to changes in vessel noise, and it is possible that habituation occurred. However, some findings indicate that harbor porpoise do not exhibit habituation to vessel noise, even after living in an environment with high levels of vessel noise [7, 16]. Therefore, further studies are necessary to comprehensively elucidate the influence of habituation to vessel noise on click parameters.

In the present study, buzz ratio was higher during the day at the two sites; additionally, there was no significant difference in the number of prey species captured in St. S and St. M. The prey species of finless porpoise [76] were similar at St. S and St. M, with the main prey species consisting of whitespotted conger (*Conger myriaster*), Japanese sea bass (*Lateolabrax japonicus*), octopus, shrimp, and squid [77–79]. Additionally, the following non-prey species of the finless porpoise were captured at both sites: Japanese jack mackerel (*Trachurus japonicus*), red tilefish (*Brachiostegus japonicus*), chubmackerel (*Scomber japonicu*), Japanese spanish mackerel (*Scomberomorus niphonicus*), largehead hairtall (*Trichiurus lepturus*), righteye flounder (Pleuronectidae sp.), and bastard halibut (*Paralichthys olivaceus*), pufferfish (Tetraodonidae sp.), and crab (Brachyura) [77–79]. Furthermore, there were no differences in depth or bottom sediments at both sites. Variations in ASL at both sites may likely not be due to differences in feeding time or prey species. However, we can not rule out the possibility that other parameters not examined in this study may have affected the behavior of finless porpoises. Therefore, further studies are necessary to comprehensively elucidate the effect of environmental factors on finless porpoises.

## Effects of day/night on echolocation click and click train parameters

In the present study, day/night had a greater impact on echolocation click and click train characteristics than vessel noise. Specifically, -3 dB BW was wider, click duration was shorter, and ICI was shorter in both sites at night. A wider bandwidth provides more information, including noise information [80]. The shorter the click duration, the higher the accuracy of binaural

time measurements, resulting in improved localization ability. Atlantic bottlenose dolphins localize using interaural differences in the arrival time of sound, binaural phase differences due to each ear being at a different point in the phase angle, or binaural intensity differences [81]. The shorter ICI observed in the present study indicated that finless porpoises scan their surrounding environment more frequently per time. During darkness, one captive harbor porpoise increased the number of echolocation click trains emitted per unit time [82]. The variations in ICI observed in the present study were consistent with a previous finding [82], and it is possible that finless porpoise exhibit a higher searching intensity of their surrounding environment during the night than during the day. Based on these findings, it is probable that finless porpoises rely more on acoustic information at night owing to relatively lower visual information at night. Therefore, the increase in bandwidth, the decrease in click duration, and shorter ICI are necessary to improve localization accuracy and information acquisition to compensate for low visual information at night.

In the present study, there was no significant difference in peak frequency and center frequency between day and night. Although the center frequency of melon-headed whale has been shown to increase at night [18], the frequency bands emitted by melon-headed whale and finless porpoise are different. Specifically, finless porpoises emit echolocation clicks at frequencies of 125–135 kHz [21, 29–31], whereas melon-headed whales produce echolocation clicks at frequencies of 25–30 kHz [83], which is a difference of approximately 100 kHz. Therefore, the fluctuation of frequency changes caused by surrounding environment of finless porpoises and melon-headed whales could be attributed to differences in frequency band. Additionally, a previous study indicated that narrow-band high-frequency species are less likely to exhibit fluctuations in frequency [84]. The range of fluctuation in frequencies in the present study was small, necessitating further studies with more data for narrow-band high-frequency species.

### Comparison of echolocation click and click train parameters with previous findings

Compared with the click and click train parameters of the finless porpoise in the Liao-dong wan Bay, Bohai Sea [29], and the Taiwan strait [42], finless porpoise examined in the present study had a higher peak frequency, wider bandwidth, and shorter click duration [29, 42]. These differences were more significant than those between St. S and St. M and may be caused by factors such as skull morphology [43, 45], environmental characteristics, and behavior. In some bat species, skull morphology has been suggested to be associated with echolocation parameters [85, 86]. Moreover, the skull morphology of harbor porpoise exhibits evolutional adaptation to prey species [87]. However, it is unclear whether there are differences in skull morphology between finless porpoises in Liao-dong wan bay, Taiwan strait, and the Japanese coastal area. Therefore, further studies are necessary to elucidate the relationship between skull morphology, click and click train parameters, and environmental factors. Harbor porpoise exhibited variations in click train parameters for communication [88]. Thus, it is necessary to examine whether variations in click and click train parameters are dependent on the behavioral state of finless porpoise.

### Conclusions

In the present study, echolocation click and click train parameters were influenced by the day/night. ASL was higher in the absence of vessel noise during the daytime at the site with low vessel traffic. Additionally, finless porpoise increased their resolution and amount of sound information by shortening the click duration, increasing -3 dB BW, and reducing ICI at night. Overall, these findings contribute to our understanding of species adaptation in response to

day/night change. To accurately assess the impact of vessel noise on echolocation characteristics, it is important to consider day/night factors. However, only acoustic monitoring was performed in this study, and factors, such as the behavioral state of vocalizing individuals and the speed and size of vessels were not considered, indicating the need for further studies on the effects of these factors on echolocation.

## Supporting information

**S1 Dataset. Click, click train, and environmental parameters for each on-axis echolocation click recorded at Seto Inland Sea (St. S), Japan.**
(XLSX)

**S2 Dataset. Click, click train, and environmental parameters for each on-axis echolocation click recorded at Mikawa Bay (St. M), Japan.**
(XLSX)

**S1 Table. Results of the best fitting GLMs analysis for several echolocation characteristics.** The explanatory variables were listed in descending order of the absolute values of their estimates, except for Intercept.
(DOCX)

## Acknowledgments

Data acquisition for St. M and St. S was supported by Tsunemi Suzuki, Kengo Ueda, Tetsuya Kohama, and Dr. Shinichi Watanabe. We also thank Dr. Tomonari Akamatsu and the staff at the Fisheries and Environment Oceanography Laboratory at Kyoto University for their support and cooperation. In particular, we would like to thank Dr. Junichi Takagi, Dr. Manabu Kume, Hirotaka Tajima, and Haruka Nakajin for data acquisition and useful discussions. Finally, we would like to thank Editage for editing and reviewing this manuscript.

## Author Contributions

**Conceptualization:** Mayu Ogawa, Satoko S. Kimura.

**Data curation:** Mayu Ogawa, Satoko S. Kimura.

**Formal analysis:** Mayu Ogawa.

**Funding acquisition:** Mayu Ogawa, Satoko S. Kimura.

**Investigation:** Mayu Ogawa, Satoko S. Kimura.

**Methodology:** Mayu Ogawa, Satoko S. Kimura.

**Project administration:** Mayu Ogawa, Satoko S. Kimura.

**Resources:** Mayu Ogawa, Satoko S. Kimura.

**Software:** Mayu Ogawa.

**Supervision:** Satoko S. Kimura.

**Validation:** Mayu Ogawa, Satoko S. Kimura.

**Visualization:** Mayu Ogawa.

**Writing – original draft:** Mayu Ogawa.

**Writing – review & editing:** Mayu Ogawa, Satoko S. Kimura.

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
