## [Decision Letter · Decision Letter 0]

29 Jun 2022

PONE-D-22-11363Changes in finless porpoise echolocation clicks in response to diel and vessel traffic presencePLOS ONE

Dear Dr. Ogawa,

Thank you for submitting your manuscript to PLOS ONE. After careful consideration, we feel that it has merit but does not fully meet PLOS ONE’s publication criteria as it currently stands. Therefore, we invite you to submit a revised version of the manuscript that addresses the points raised during the review process.

In particular, authors should address the comments on statistical analysis, such as handling confounding effects and interpreting results. In addition, there is room for improvement in the expression of sentences and organization of the text.

We look forward to receiving your revised manuscript.

Kind regards,

Kentaro Q. Sakamoto, Ph.D., DVM

Academic Editor

PLOS ONE

Journal Requirements:

2. We note that Figure 1 in your submission contain copyrighted images. All PLOS content is published under the Creative Commons Attribution License (CC BY 4.0), which means that the manuscript, images, and Supporting Information files will be freely available online, and any third party is permitted to access, download, copy, distribute, and use these materials in any way, even commercially, with proper attribution. For more information, see our copyright guidelines: http://journals.plos.org/plosone/s/licenses-and-copyright.

1. You may seek permission from the original copyright holder of Figure 1

to publish the content specifically under the CC BY 4.0 license. 

Reviewers' comments:

Reviewer's Responses to Questions

**Comments to the Author**

1. Is the manuscript technically sound, and do the data support the conclusions?

Reviewer #1: Partly

Reviewer #2: Partly

2. Has the statistical analysis been performed appropriately and rigorously? 

Reviewer #1: No

Reviewer #2: Yes

3. Have the authors made all data underlying the findings in their manuscript fully available?

Reviewer #1: Yes

Reviewer #2: Yes

4. Is the manuscript presented in an intelligible fashion and written in standard English?

Reviewer #1: Yes

Reviewer #2: No

5. Review Comments to the Author

Reviewer #1: I read the manuscript with great interest as it contains a combination of basic understanding on echolocation of a species with red listed sub-species, and impacts of the increasing global anthropogenic threat posed by vessel noise.

The authors have recorded on- and off-axis echolocation clicks of finless porpoises and underwater sounds at two sites during approximately two months. The two sites have different number of fishing vessels in nearby harbours, and one of the sites is close to several ferry lines. In on-axis data, seven click or click train parameters are analysed and compared between the sites, between day and night, and between vessel noise absence and presence. Correlation between the on-axis click parameters are investigated. In on- and off-axis data, the total number of regular clicks and buzz-clicks are calculated per hour of day.

The manuscript is in general easy to read, although it can be improved. The data-collection and the acoustic analyses are carried out well, but I have questions on the statistical analyses of the data, and how much of the results that actually can be related to vessel (noise) impact. The results of the study are of importance for increased understanding on odontocete echolocation behaviour, impact of vessel noise, and conservation of threatened species.

Major comments:

• As I understand how the statistical analyses were carried out, there may be a confounding effect between day/night and vessel noise absence/presence. This is fundamental for the results and their interpretation.

• In addition to the confounding effects, there is little data supporting the conclusions on that the differences between the two sites actually is due to vessel (noise) presence. How do we know that it is not due to several other environmental factors that may vary between the sites, such as prey composition, quality and quantity? These conclusions would need a stronger basis.

Minor comments:

• The abstract need further information for the reader to understand the work carried out.

• I find the first part of the Introduction a bit unclear on what you want to present and why.

• I suggest additional references that may be useful (otherwise disregard them), and wonder about the links leading to web pages in Japanese.

• The terminology of the acoustic parameters needs to be clearer (I suggest to use “click duration” instead of “t”, “clicks per train” instead of “pulses” etc), and it needs to be clarified which parameters that are click parameters and which are train parameters.

• I think the definition of buzz differs from what often is used, please clarify and discuss how the definition used may impact the results.

• I do not understand how the definition of day and night relates to the light conditions (dusk and dawn).

• The number of samples in the different categories (Site S/M, day/night and vessel noise absence/presence) need to be presented more clearly.

• In Results, it would be easier to follow if the results were presented in the same order as Fig. 3.

• As the frequency of the clicks of narrow-band high-frequency species varies extremely little, is not click duration and number of cycles per click basically two different measurements of the same thing? Are both needed?

• Centre frequency seems to be missing in S1 Table.

• Have you considered correcting the p-value for multiple comparisons of the same data?

• It is not clear why the rate of all clicks and buzzes is presented as hourly histograms and not as box plots as the on-axis click/train parameters, and there is no explanation to why all clicks and buzzes were not analysed in relation to vessel noise absence/presence. It may have been out of the scope of the study, or technical constraints, but it leaves me wondering.

• The sub-headings in the Discussion are helpful. Consider if the manuscript can be arranged in the same general order throughout.

• Overall the Discussion is supported by the Results as they are now, but need to be revised if the Results are revised (if the statistical analyses are revised).

• The Discussion revolves around the energy needed for echolocation, which is relevant, but I miss further comments on the ecological implications of the findings. May prey differences between the sites explain some of the results? How does increased click duration improve echolocation during vessel noise presence?

• The Conclusions are a summary of the findings. I think it would be improved by shortening it, highlighting only the most important findings.

• The tables and figures are overall clear, details are commented.

Please see commented manuscript for detailed comments.

Reviewer #2: Review of “Changes in finless porpoise echolocation clicks in response to diel and vessel traffic presence”

I found this manuscript hard to follow because of grammatical issues. Many grammar mistakes make the article unclear and a grammar check by a native English speaker is definitely required. I mentioned some of the mistakes in my comments, but these are just examples, and the full paper should be corrected.

In the manuscript, the gap in the literature is unclear. As a reader, I am not able to understand what has been done on this species and other odontocetes in previous studies and therefore what is new here (except the location).

The material and methods part is very hard to follow because of grammar, repetitions, and bad organization.

Discussion overall contains good hypotheses, except for the difference between the two sites that focus on some potential explanations and forget some others. Moreover, some studies conducted on captive porpoises (asiaeorientalis and sunameri) were never cited while they could be useful to interpret some of your results.

In addition, no limitations are mentioned. The dataset is very small, and this should be mentioned, especially since the results given are surprising.

Title: is it grammarly correct to say “in response to diel”? I believe it is not, better find another wording.

Abstract:

L16: navigate in their environment

L17: “making it… an important role” is not correct grammatically speaking

L22: where?

L22: “the effects of diel response” is not grammatically correct

L24: diel cycle of visitation? Feeding sound rate? What does this mean?

L25: responses were not constant for the same species? You actually studied a single species, so no need to mention this. And what does “constant” mean?

L27: per click train

L27-28: during the minute

L29-30: bad wording. I suggest “In contrast, in the area subject to more boat traffic, click length and number of cycles per click decreased and source-level increased before and after vessel traffic noise.”

Introduction:

L40: their characteristics (no need to repeat “echolocation clicks”)

L44: I don’t think you can speak of acoustic observations since it is not something visible. Use “recording” instead.

L43-45 and 46-48: these two sentences are saying opposite things… you tell us that most studies did not study diel rhythm and then you state that diel rhythm has been studied a lot through PAM…

L54-56: you just gave examples that say exactly this… I suggest removing this sentence or placing it before the examples.

L65: you first speak about the effect on vocalizations and then give an example about energetic budget and then speak about vocalizations again… I cannot see the usefulness of this example here.

L72: is necessary

L74-83: you are mixing information on two subspecies. If your goal is to state that there has been research on this topic for asiaeorientalis but not for sunameri, state it clearly and don’t mix information between these two subspecies.

L88-91: it would be interesting to know what the other studies of clicks on the same species found, even if it is in different locations.

L86-91: this information belongs to the previous paragraph, it should be given before you state what you did in your study.

Matériel and methods:

L97: acoustic recording

L101: boats

L124-125: a total of six days of recording for each area? That is not what I understand from table 1…

L128: was not able

L130: in the click train of the echolocation? What does it mean?

L134: was used to discriminate?

L126-139: overall very unclear, and hard to understand, mostly because of the bad grammar.

After reading this part, it is not clear to me how the acoustic data was collected. The collection period is around 2 months in table one, for a total of 638h for the first location for instance, and the text mentions 6 days of data. So, how was the data sampled? Was it 1-minute recording each hour for example?

L160: you should say that “a wave with an interval of less than 1ms from the previous was discarded…”

L165: what exactly was checked?

L170: remove “was”

L172: click trains with constant amplitudes or other amplitude patterns were not included?

L173: the (4) does not give any criteria, what should the maximum amplitude be like?

L176-178: what about criteria 4 and 5?

L197: salinity data was obtained from temperature data?

L201-203: a good example of an explanation that is very unclear because of grammar. You should write: “The ICI was

the time interval between consecutive pulses of a click train.” I don’t understand if you speak about the amplitude of a click or the difference of amplitude between two clicks, therefore I cannot even reformulate your full sentence…

L228: produced by porpoises

L231: why is the day/night separation fixed like this when you stated earlier that you used the sunset/sunrise data? It is not consistent…

Results:

L234-235: only that little clicks? Are you sure you don’t mean click trains? Some click trains contain more clicks than this…

Table 2: you need to provide details for the stat tests, for example, the exact p-value and the U for Mann Whitney.

L269-281: why do we have a table for inter-site differences, but everything is written in the text for other differences? I recommend staying consistent in the way you present the same kind of results.

I recommend being careful with the words increase and decrease. You can say that ASL was higher after a boat noise for instance, but saying that ASL increases after a boat pass by has a more specific meaning, thy does not correspond to the analysis you did.

L106-313: no stats here? You did not analyze the buzz ratio differences between night and day?

Discussion:

L319: of on-axis

L325: parameters

L326: click trains and buzzes

L327: don’t refer to figures in the discussion

L327: is it a higher presence or a higher click emission? Porpoises could be there and emit much fewer clicks because of some diurnal changes in behavior or because of human activities.

L330: same

L333: why use “although” here? There are no contradictions between the two parts of the sentence…

L343-344: which changes? Same as yours?

L344-345: I am not currently reading a lot about this topic, but it surprises me a lot that only one study investigated diel changes in clicks in odontocetes… I believe there are studies on Yangtze finless porpoises at least, and probably on bottlenose dolphins too. I suggest that you make sure you cited all the relevant literature.

L358: remove “exhibit”

L350-357: here, you cite some literature about diel changes in odontocetes, but in the previous paragraph, you say that there are almost no studies about this… Please be clear on what points are new in your study and have not been investigated yet, because for now, it looks like you overstate the novelty of your study.

L365: suggest

L364-369: why is this information given separately from the other paragraphs? The argument is the same…

L388: how can an increase in vigilance result in a lower ASL? Would they consider boats as predators and therefore be careful not to be detected by this potential predator? Or do you have another hypothesis? This result is really weird compared to previous studies. if true, it would mean that finless porpoises have a different strategy against noise than other odontocetes and that they may be impacted more since it is impacting their communication and echolocation much more. I would like a more detailed discussion on this point. I would also like you to discuss a bit the habituation process.

L389-390: it would have been interesting to analyze clicks after the noise to see when parameters come back to “normal”

L391-395: such a big difference between the two populations. If these data were mine, I would

L406: habituate is a verb, not a noun, you cannot use it like this. Check elsewhere

L429: present study

L372-441: discussion elements are present, but like elsewhere, some problems with grammar and organization make it hard to follow.

L446-449: we are in the part about the difference between sites, why mentioning again day and noise effects? Keep this for the conclusion.

L455-457: or they may have different feeding strategies!

L457-458: what does it mean? Were porpoises visually observed during all the recordings? I don’t think so. The only thing that could explain this is habitat use: maybe you put your array in a feeding ground for a site and in a traveling area for the other site. But there are many other hypotheses like the different feeding behavior, or the different prey density.

Figure 1: I suggest adding the name of each instrument to the D figure.

Figure 5: Y-axis title missing

6. PLOS authors have the option to publish the peer review history of their article (what does this mean?). If published, this will include your full peer review and any attached files.

Reviewer #1: No

Reviewer #2: No

---

## [Author Response · Author response to Decision Letter 0]

28 Apr 2023

We thank you and the reviewers for your thoughtful suggestions and insights. According to your comments, we completed to revise the manuscript. The manuscript has benefited from these insightful suggestions. I look forward to working with you and the reviewers to move this manuscript closer to publication in the PLOS ONE.

---

## [Decision Letter · Decision Letter 1]

13 Jun 2023

PONE-D-22-11363R1Variation in echolocation click characteristics of finless porpoise in response to absence/presence vessel noise and day/nightPLOS ONE

Dear Dr. Ogawa,

Thank you for submitting your manuscript to PLOS ONE. After careful consideration, we feel that it has merit but does not fully meet PLOS ONE’s publication criteria as it currently stands. Therefore, we invite you to submit a revised version of the manuscript that addresses the points raised during the review process.

The manuscript has been greatly improved. However, a little more work is required.

I agree with reviewer 3’ comment that concerns about statistical analysis need to be addressed. Prior to using GLM, you examined the differences of parameter values between the sites by t-test or U-test. However, these datasets can be biased in the sampling process (e.g. one site may have more samples taken during the day, while another site may have fewer samples taken at night). In such cases, simple t-test and U-test are not suitable for examining differences. I suggest using GLMM instead of GLM for all statistical models. Alternatively, as reviewer 3 suggested, you could apply GLM including “study site” as an additional explanatory variable.

In discussion, you can mention about the difference between the study sites (especially with regard to Fig. 3A). But since you did not examine the effect of the sites by statistical tests, you should tone down the message (line 356-358, 369-370 and maybe 456-457). The use of “significantly” is not suitable.

In this paper, vessel noise did not significantly affect echolocation click and click train parameters (according to the abstract). If so, you should tone down the phrase “absence/presence of vessel noise” in the title. The possible title could be “Variations in echolocation click characteristics of finless porpoise in response to day/night and absence/presence of vessel noise”.

We look forward to receiving your revised manuscript.

Kind regards,

Kentaro Q. Sakamoto, Ph.D., DVM

Academic Editor

PLOS ONE

Journal Requirements:

Reviewers' comments:

Reviewer's Responses to Questions

**Comments to the Author**

1. If the authors have adequately addressed your comments raised in a previous round of review and you feel that this manuscript is now acceptable for publication, you may indicate that here to bypass the “Comments to the Author” section, enter your conflict of interest statement in the “Confidential to Editor” section, and submit your "Accept" recommendation.

Reviewer #3: (No Response)

2. Is the manuscript technically sound, and do the data support the conclusions?

Reviewer #3: Partly

3. Has the statistical analysis been performed appropriately and rigorously? 

Reviewer #3: No

4. Have the authors made all data underlying the findings in their manuscript fully available?

Reviewer #3: (No Response)

5. Is the manuscript presented in an intelligible fashion and written in standard English?

Reviewer #3: Yes

6. Review Comments to the Author

Reviewer #3: I was pleased to review PONE-D-22-11363R1 “Variation in echolocation click characteristics of finless porpoise in response to absence/presence vessel noise and day/night”. The authors ask an interesting question, and the manuscript is generally clearly written. However, I am concerned about certain aspects of the methods and the statistical analyses. As such, it is difficult to ascertain whether the interpretation of the results is in fact supported by the data.

Specific comments:

P12

L240–246: “Click and click train ~, and environmental parameters,”

The authors mix information theory (i.e., model selection with an information criterion – in this case AIC) with null hypothesis testing, which is not recommended. I commend the authors to simply use GLMs (which include "study sites" as an additional explanatory variable) for all the click parameters, without testing beforehand. In addition, please note that the relative importance of model parameters and support for models is meaningless unless the authors provide a measure of the variance (or deviance) explained.

L252–253: “...the site (factor type) was the random variable in GLMM.”

When you include "sites" as a random variable in GLMM, you cannot compare any results between study sites (because it means that you decided to regard difference between sites as a random effect, not a fixed effect). If the authors want to compare and discuss about site difference (which I think the authors do), using GLMs that include "sites" as an explanatory variable may be suitable.

L253: “Salinity data~”

First appearance of "salinity" in the manuscript. Where do they come from? As there is no description about salinity data in "Environmental data" section above (and the authors do not use this data after all), please delete this sentence.

P14

L284–285: “..., and environmental parameters~”

I suggest the authors to change the listing order of the parameters in the table so that background information (temp, flow, noise) comes first and data on behavioral response (i.e., click parameters) comes second. (I think listing in such order makes this table more easy-to-read for readers.)

P15

L303–304: “...with Residual Maximum Likelihood estimation (REML).”

This information should appear first in Methods section.

L305: “GLM with gamma family~”

Information about families of distribution (as well as the link function) used for GLMs should be mentioned in Methods section, not in Results. (Note that this comment also applies to other paragraphs within this section.)

P18

L350–353: “To the best of our knowledge~”

This sentence is not a discussion; it is just stating what the author did. This should be removed. Just starting your discussion from the first section "Effects of presence~" seems enough.

L357–358: “..., which was contrary to previous findings.”

This statement is untrue because (, as the authors mentioned in Lines 361-364,) there are several studies which showed decreases of ASL in the presence of vessels. This should be toned down.

P19

L376–377: “... at the site with high [or low] vessel noise (St. M [or St. S])”

Although the difference in the noise level between two sites is significant (i.e., p<0.05), its effect size seems to be very small (116±1 in St. S vs. 118±2 in St. M). I think the authors should reconsider the importance and meaning of this subtle difference. In other words, "can this subtle difference in noise level (only 1 dB!) really affect behaviors of finless porpoise?"

L385: “..., there was no significant difference in the number of prey species~”

I could not find any details about sampling methods, statistics, and analyses for data regarding prey organisms. Please add this information to Methods section if the authors want to present these results here. Also, there seems a difference in the number of species between St. S (two fish species plus octopus, shrimp, squid) and St. M (eight fish species plus octopus, shrimp, squid). Please re-check.

Besides, what kind of shrimp and squid were caught? Please identify these organisms to species level (or, at least, to a higher taxonomic level; order, class, family, etc.).

L386–398: “The following prey species~”

Why listing up all the species here? This section does not seem to be related to any of your discussion. Are these species diurnal or nocturnal? Which of them are important for finless porpoise? Just delete this section or add more context (if there is any importance in listing up each of prey species here).

P20

L400–401: “... noise levels between the two sites were different~”

As I commented above (L376–377), the difference in noise level (i.e., 1 dB) do not seem to be great enough to cause any change in porpoise's behavior. This point needs to be considered.

L414: “... by finless porpoise.”

This phrase is redundant. Please delete.

L419–420: “Finless porpoises rely more on~”

Does this statement come from your observation (i.e., data presented in this manuscript) or other literature? If the latter, please provide reference.

P21

L432, L434: “NBHF”

First appearance of "NBHF" in the manuscript. Abbreviations must be defined upon first use. Here, the authors better use "narrow-band high-frequency" instead of "NBHF" because the authors use this abbreviation only twice in the whole manuscript and I think most readers are not familiar with this abbreviation.

L436: “In this present study, ~”

This is not a discussion. This is what the authors did. Please delete.

L439–441: “... in the present study had a higher peak frequency~”

Here the authors compare four sites (two in Japan, Liao-dong wan Bay, and Taiwan strait) but only show data from two sites abroad. I think is better to delete numerical values from this sentence for simplicity. If these numerical values are important, showing numerical values for two Japanese sites might be helpful for readers to compare among four sites. In this case, providing another table (that compares values from four study sites) might be helpful.

7. PLOS authors have the option to publish the peer review history of their article (what does this mean?). If published, this will include your full peer review and any attached files.

Reviewer #3: No

---

## [Author Response · Author response to Decision Letter 1]

23 Jun 2023

We sincerely appreciate the valuable suggestions and insightful feedback provided by editor and the reviewers. The manuscript has greatly benefited from these constructive inputs. These revisions have resulted in significant enhancements to the content and quality of the manuscript, enabling us to present clearer and more reliable results. I look forward to working with you and the reviewers to move this manuscript closer to publication in PLOS ONE.

---

## [Editor Report · Decision Letter 2]

27 Jun 2023

PONE-D-22-11363R2Variations in echolocation click characteristics of finless porpoise in response to day/night and absence/presence of vessel noisePLOS ONE

Dear Dr. Ogawa,

Thank you for submitting your manuscript to PLOS ONE. After careful consideration, we feel that it has merit but does not fully meet PLOS ONE’s publication criteria as it currently stands. Therefore, we invite you to submit a revised version of the manuscript that addresses the points raised during the review process.

Do not remove the t-test and u-test descriptions from the Materials and Methods section. You are looking at significant differences using these tests in Results and Table 2. Additionally, I think it would be better to switch the order of Table2 and 3.

We look forward to receiving your revised manuscript.

Kind regards,

Kentaro Q. Sakamoto, Ph.D., DVM

Academic Editor

PLOS ONE
---

## [Author Response · Author response to Decision Letter 2]

28 Jun 2023

We sincerely appreciate the valuable suggestions and insightful feedback provided by the editor. The manuscript has greatly benefited from these constructive inputs. These revisions have resulted in significant enhancements to the content and quality of the manuscript, enabling us to present clearer and more reliable results. Thank you for your kind consideration. I look forward to hearing from you.

---

## [Editor Report · Decision Letter 3]

29 Jun 2023

Variations in echolocation click characteristics of finless porpoise in response to day/night and absence/presence of vessel noise

PONE-D-22-11363R3

Dear Dr. Ogawa,

We’re pleased to inform you that your manuscript has been judged scientifically suitable for publication and will be formally accepted for publication once it meets all outstanding technical requirements.

Kind regards,

Kentaro Q. Sakamoto, Ph.D., DVM

Academic Editor

PLOS ONE
---

## [Editor Report · Acceptance letter]

27 Jul 2023

PONE-D-22-11363R3 

Variations in echolocation click characteristics of finless porpoise in response to day/night and absence/presence of vessel noise 

Dear Dr. Ogawa:

I'm pleased to inform you that your manuscript has been deemed suitable for publication in PLOS ONE. Congratulations! Your manuscript is now with our production department. 

Kind regards, 

on behalf of

Dr. Kentaro Q. Sakamoto 

Academic Editor

PLOS ONE